# Deregulating the CYP2C19/Epoxy-Eicosatrienoic Acid-Associated FABP4/FABP5 Signaling Network as a Therapeutic Approach for Metastatic Triple-Negative Breast Cancer

**DOI:** 10.3390/cancers12010199

**Published:** 2020-01-13

**Authors:** Maria Karmella Apaya, Pei-Wen Hsiao, Yu-Chih Yang, Lie-Fen Shyur

**Affiliations:** 1Molecular and Biological Agricultural Sciences Program, Taiwan International Graduate Program, Academia Sinica, Taipei 115, Taiwan; mlapaya@gate.sinica.edu.tw; 2Agricultural Biotechnology Research Center, Academia Sinica, Taipei 115, Taiwan; pwhsiao@gate.sinica.edu.tw (P.-W.H.); ycyang12@gate.sinica.edu.tw (Y.-C.Y.); 3Graduate Institute of Biotechnology, National Chung Hsing University, Taichung 402, Taiwan; 4Biotechnology Center, National Chung Hsing University, Taichung 402, Taiwan; 5PhD Program in Translational Medicine, College of Medicine, Kaohsiung Medical University, Kaohsiung 807, Taiwan

**Keywords:** triple-negative breast cancer, tumor relapse, metastasis, fatty acid binding protein, phytogalactolipid, doxorubicin

## Abstract

Recurrence and metastasis are the main causes of triple-negative breast cancer (TNBC) mortality. On the basis of our clinical cohorts and integrative omics analyses, we hypothesized that understanding the interplay between fatty acid binding protein (FABP) and epoxy-eicosatrienoic acid (EET) driven metastatic progression can uncover a new opportunity for TNBC intervention. In this study, the biological relevance of increased protein expression of CYP2C19, FABP4, and FABP5 in TNBC tumors and in the TNBC cell line (MDA-MB-231), as well as its highly metastatic lung seeking variant (LM6) were delineated from publicly available datasets, shRNA-mediated knockdown, EET supplementation, cancer and stromal cell co-cultures, and an orthotopic and resection xenograft tumor mouse model. We found that the high expression levels of CYP2C19 and FABP4 and FABP5 are critical in TNBC metastatic transformation and stromal cell interactions. Furthermore, EET-associated nuclear translocation of FABP4 and FABP5 and nuclear accumulation of SREBP-2 or PPAR-γ influence TNBC cell proliferation, migratory transformation, and distal metastasis priming. Most notably, we uncovered novel bioefficacy and modes of action of the anticancer drug doxorubicin and a phytogalactolipid, 1,2-di-*O*-α-linolenoyl-3-*O*-β-galactopyranosyl-*sn*-glycerol (dLGG), which effectively attenuated TNBC recurrence and lung metastasis through deregulating the FABP/EET dynamics and levels. This study, therefore, introduces a novel approach to combating TNBC by targeting the FABP/EET/CYP-associated metastatic signaling network.

## 1. Introduction

Triple-negative breast cancer (TNBC) accounts for 15% to 20% of the incidence of global breast cancer but is disproportionally associated with breast cancer-related death [1]. Chemotherapy, the sole therapeutic route for TNBC patients, causes severe toxicity, resistance, and relapse; therefore, a better understanding of TNBC biology is needed. Mechanistic studies show strong links between adiposity, fatty acid metabolism, and BC risk [2,3,4,5], establishing a rationale for exploring related networks such as TNBC targets. Interest in the potential of the fatty acid binding protein (FABP) family, in particular, has recently been gaining traction. The nine known FABP isoforms have specific cellular and tissue localization and ligand binding preferences through which they regulate the uptake, storage, and metabolism of lipophilic compounds [6,7]. They activate transcription factors that are associated with proliferation, differentiation, and vascularization in multiple cancer types [8,9,10]. They may also act as mediators of cancer cell-stromal interactions [3,11,12]. 

FABPs have varied functions in BC progression. FABP3 is a tumor suppressor, whereas FABP4, FABP5, and FABP7 are associated with poor survival [13,14,15,16]. Among the FABP isoforms, FABP4 and FABP5 likely play critical roles in mammary tumor development [17,18,19,20]. FABP5 is linked to breast cancer proliferation and resistance, whereas serum and tumor FABP4 levels were found to be significantly higher in BC patients than that of healthy controls and were positively connected with tumor size, aggressiveness and lymph node involvement [14,21,22,23,24]. Few studies, however, have investigated the regulation of FABPs in TNBC. To date, the functions of FABP4 and FABP5 are incompletely understood, and their roles in hormone independent TNBC signaling are yet to be deciphered. 

We previously showed that arachidonic acid-derived EETs are important metastasis drivers in TNBC tumors. Elevated EET concentrations are associated with co-upregulation of CYP450 epoxygenases CYP2C19 and CYP2J2, FABP4 and FABP5, and adipocyte signaling-related proteins in TNBC tumors [25]. EETs are also important in epithelial-mesenchymal transition (EMT) and resistance via the STAT and AKT signaling pathways [26,27,28,29,30]. From these observations, we hypothesized that FABP4 and FABP5 have dynamic roles in EET-mediated TNBC signaling, particularly in the activation of pathways involved in stromal interaction and metastasis transformation. 

In this study, we derived the clinical relevance of FABP4 and FABP5 upregulation from publicly available datasets and used in vitro metabolite supplementation and co-culture systems and an in vivo xenograft orthotopic and resection mouse model to analyze their roles in TNBC tumor relapse and metastasis. We highlight the therapeutic potential of a plant-derived galactolipid, 1,2-di-*O*-α-linolenoyl-3-*O*-β-galactopyranosyl-*sn*-glycerol (dLGG), previously shown to have anti-melanoma metastasis and immunomodulatory activities via lipid-associated mechanisms [31,32,33], in regulating the FABP/EET-mediated signaling axes in TNBC. These findings illustrate the distinct biological activities of FABP4 and FABP5 in this disease and provide a rationale for the regulation of these two isoforms as a strategy for TNBC intervention.

## 2. Results

### 2.1. Translational Implications of the Upregulation of CYP2C19/FABP4/FABP5 Signaling Axis in TNBC Tumors

We hypothesized that FABP4 and FABP5 are critical nodes for EET-driven TNBC oncogenic signaling because these lipid chaperone proteins are upregulated in metastatic TNBC tumor specimens overexpressing CYP epoxygenases (CYP2C19) [25]. Linear regression analysis of transcriptomics data derived from the The Cancer Genome Atlas (TCGA) [25] showed a positive correlation between FABP4, FABP5, and CYP2C19 gene expression in TNBC and ER+/PR+BC (Figure 1a). Relapse-free survival (%RFS) indicated that FABP4 (*p* = 0.015) and FABP5 expression (*p* = 0.075) correlated with TNBC occurrence and decreased survival, but not in hormone positive mammary tumors (Figure 1b). Approximately 19% to 25% of metastatic TNBC tumors (*n* = 62) and 9% to 18% of total TNBC tumors (*n* = 160) had upregulation of these three genes in two independent, non-overlapping TCGA cohorts (Figure 1c). Further analysis of the associated gene network in TNBC tumors co-overexpressing FABP4/FABP5/CYP epoxygenase (mRNA *z*-score ≥ 2.0) showed enrichment of key genes involved in lipid metabolism, metastasis transformation, and stromal interaction, including SREBF1, ALDH3A1, Src, MMPs, PPARs, CD44, and chemokine- and interleukin-associated proteins (CXCRs) (Figure 1d). These genes are associated with the top 10 signaling pathways most significantly upregulated in metastatic TNBC tumors (Appendix A), suggesting that they are important in migratory and invasive transformation. Results from this in silico analysis highlight that intercepting the CYP2C19/FABP4/FABP5 signaling network could provide a new therapeutic opportunity for a subpopulation of TNBC patients. 

### 2.2. In Vitro Functional Analysis: CYP2C19/FABP4/FABP5 Are Intrinsically Increased in Lung-Seeking TNBC Cells and Functionally Associated with EET-Mediated Metastasis Transformation 

On the basis of our previous finding that intrinsic CYP epoxygenase upregulation and elevation of EET metabolites are more pronounced in mesenchymal-like TNBC cells (e.g., MDA-MB-231) as compared with immortalized mammary epithelial cells (MCF10A), basal-like TNBC (e.g., MDA-MB-468 and HCC 1937) or hormone receptor positive (e.g., MCF7 and SKBR3) cell lines [25], in this study, we focused on assessing the functional roles of this signaling axis in the metastatic transformation of MDA-MB-231 TNBC cell line and its highly metastatic lung-seeking subclone. We utilized MDA-MB-231 cells with a dual reporter system (designated 231-iR2L) and their highly metastatic lung-seeking variant (designated LM6) for in vivo and in vitro studies. We confirmed that the expression of FABP4 and FABP5 was upregulated in LM6 cells as compared with a surrogate cell line representing immortalized mammary epithelial cells (MCF10A), the parental MDA-MB-231, MDA-MB-231-iR2L, and earlier less metastatic clones LM2 and LM4 (Figure 2a). The protein expression levels of CYP2C19, FABP4 and FABP5 were significantly increased in LM6 cells as compared with 231-iR2L and earlier LM sublines, alongside the proteins identified in the in silico network analysis, namely EMT (RhoA and vimentin), metastasis (p-Src^419^/Src and p-FAK/FAK), stromal interaction (MMP-9), and stem cell-related markers (CD44 and ezrin), (Figure 1d). Representative Western blots from three independent experiments are shown in Figure 2b. Corresponding statistical and densitometry ratio analyses are presented in Appendix A. These results suggest that the identified protein network is interrelated in EET-driven metastatic TNBC signaling. 

We used shRNA knockdown to investigate whether the increased expression of CYP1C19, FABP4, and FABP5 in LM6 contributed to its metastatic nature. We depleted these proteins in LM6 using specific shRNA constructs, creating LM6-shFABP4, LM6-shFABP5, and LM6-shCYP2C19 cell clones. The shRNA clones with more than 50% depletion of target proteins were used in subsequent experiments (Figure 2c). Doubling times and proliferative phenotypes were similar among all the shRNA knockdown clones and control cells. Interestingly, decreased protein levels of CYP2C19 were observed in both FABP4 (LM6-shFABP4-1) and FABP5 (LM6-shFABP5-2) knockdown cells, and vice versa. Protein expression levels of both FABP4 and FABP5 were downregulated in CYP2C19-depleted cells (LM6-shCYP2C9-2), indicative of a potential feedback regulation mechanism along the FABP/EET signaling axis. The shRNA depletion of any of these three proteins resulted in a modest to a significantly decreased expression of EMT (vimentin and RhoA), metastasis (phosphorylation of Src and FAK), and stem-cell-related (CD44 and ezrin) markers. Representative Western blots from three independent experiments are shown in Figure 2c. Corresponding statistical and densitometry ratio analyses are presented in Appendix A. While all these markers were significantly decreased in CYP2C19-depleted cells, a more pronounced decrease in CD44 and vimentin expression was observed in the LM6-shFABP4 cells. Remarkably, FABP5-depletion led to the most significant increase in the inactive form of Src (increased p-Src^527^). These results indicate that CYP2C19, FABP4, and FABP5 are involved in the regulation of divergent signaling cascades ultimately leading to the metastatic phenotype of TNBC cells.

To determine whether intracellular EET levels are affected by CYP2C19/FABP4/FABP5 gene knockdown, we measured the basal EETs metabolite levels in quiescent parental (231 and 231-iR2L) and metastatic (LM6) TNBC cells, and in the CYP2C19/FABP4/FABP5-depleted clones (Figure 2d). Basal concentrations of all EET isomers were elevated in LM6 and LM6-shLacZ as compared with 231 and 231-iR2L. Relative to LM6 or LM6-shLacZ, the EET levels were significantly lower in LM6-CYP2C19 (all EET isomers), LM6-shFABP4 (5,6-, 11,12- and 14,15-EET isomers), and LM6-shFABP5 (5,6-, 11,12- and 14,15-EET isomers) cells. We then measured the intracellular EET levels following 24 h culture in media supplemented with 10 nM EETs (containing the four EET isomers, 2.5 nM each) to investigate the effects of CYP2C19/FABP4/FABP5 gene/protein knockdown on exogenous EET uptake. Following EET supplementation, levels of the four EET isomers in LM6 cells and LM6-shLacz (Figure 2e) cells were similar to the intrinsic EET levels detected in the parental 231 cells (Figure 2d) with the levels falling within 500 to 600 pg/10^6^ cells. These results imply that the dynamic balance of intracellular EETs in the lung-seeking TNBC cells was maintained. Following EET supplementation, intracellular concentrations of the four EET isomers in CYP2C19-depleted cells (300 to 480 pg/10^6^ cells) were significantly lower relative to LM6-shLacz cells (500 to 600 pg/10^6^ cells) (Figure 2d). These EET levels were higher than the detected intracellular EETs (220 to 275 pg/10^6^ cells) in the cells grown in culture medium without additional exogenous EETs (Figure 2d). These results suggest that the machinery for exogenous EET uptake is still partially functional in LM6 cells depleted of CYP2C19 gene/protein. Conversely, the intracellular 5,6- and 14,15-EET levels were significantly decreased in the LM6-shFABP4 clones (180 to 220 pg/10^6^ cells) (Figure 2d) and were increased to 300 to 350 pg/10^6^ cells after addition of EETs in the culture medium, while no changes on 8,9- and 11,12-EETs levels were observed (Figure 2e). In the LM6-shFABP5 cells, there was no statistical difference between the level of intracellular EETs grown with or without EET supplementation (Figure 2e). These data suggest that the four EET isomers could have differential affinity to FABP4/FABP5 and the pronounced decrease of intracellular EETs in FABP4-depleted cells implies that FABP4 could play a more important role than FABP5 in the uptake or transportation of exogenous EETs.

The roles of CYP2C19, FABP4, and FABP5 in EET-mediated metastasis were, then, examined using phenotypic Boyden chamber invasion and migration assays, and colony formation and cell motility experiments (Figure 3a–d). A significant increase in the metastatic phenotypes (migration, invasion, colony formation, and motility profiles) of 231-iR2L, LM6-shLacZ, and LM6-shCYP2C19 cells were induced following the addition of any of the four EET isomers (10 nM each) as compared with cells grown in the vehicle control (0.05% DMSO). This implied that CYP2C19 depletion does not hamper the metastatic potential of TNBC cells induced by exogenous EETs. The FABP4-depleted cells were much less metastatic than the LacZ control and CYP2C19-depleted cells. These characteristics were not changed upon EET addition. Moreover, the metastasis phenotype of FABP5-depleted cells was not changed significantly in the presence of exogenous EETs. These results indicate that deregulating the FABP/EET signaling axis can provide a means to control TNBC metastasis.

### 2.3. Potential Mechanistic Involvement of Oncogenic and Lipogenic Transcription Factors: EET-Induced Nuclear Translocation of FABP4 and FABP5 Facilitates Increased Nuclear Expression of SREBP-2 and PPAR-γ

Nuclear translocation of FABP4 and FABP5 proteins mediated by lipid ligands, including EETs, can activate transcription factors (TFs) to initiate proliferative or cell death signaling in several cancer models [35,36,37]. To explore the relevance of elevated EETs to the cellular functions of FABP4 and FABP5 in TNBC cells, we determined the nuclear expression of FABP4 and FABP5, as well as of known fatty acid (FA) activated transcription factors (TFs) such as PPARs, SREBPs, LXR, and PXR, following exogenous EETs treatment in LM6 cells (Figure 3e). Whereas the expression and cellular localization of CYP2C19 were not affected by EET treatment, on the one hand, we observed that nuclear translocation of FABP4 and FABP5 proteins was significantly induced by EETs supplementation and was accompanied by increased nuclear accumulation of PPAR-γ and SREBP-2. On the other hand, the two known FA-binding TF families, LXR and PXR, as well as SREBP-1 and PPAR isoforms α, β, and δ with intrinsically low expression in TNBC cells, were not detected by Western blotting both in the vehicle and EET-treated cells. Interestingly, RXRα was minimally detectable in the total and cytosolic fractions but was significantly detected in the nuclear fraction of the vehicle treated cells, which was attenuated following EET supplementation. The addition of EETs slightly decreased RAR protein in the total and cytoplasmic fractions while not significantly affecting its nuclear accumulation. We speculate that FABP4- and FABP5-derived EETs divert lipid-mediated signaling involved in the anti-proliferative PPAR/RXR axis [38,39] to the oncogenic cascades activated by PPAR-γ and SREBP-2 [40,41,42]. Indeed, direct protein targets of SREBP-2 (CD36, c-myc, and CD44) and PPAR-γ (c-myc and Src signaling) were increased in LM6-shLacZ control cells grown in EET-supplemented media (Figure 3f). In FABP4/FABP5-depleted cells, addition of EETs did not induce the expression of PPAR-γ and SREBP-2, as well as of their direct protein targets (Figure 3f). The corresponding statistical and densitometric ratio analyses from three independent Western blotting experiments in Figure 3e,f, are presented in Appendix A, respectively. These results suggest that elevated cellular EET levels influence FABP4 and FABP5 nuclear translocation and that could affect the subsequent expression of oncogenic and lipogenic TF signaling; placing them as key nodes in the metastatic TNBC network. 

### 2.4. Potential Mechanistic Involvement of Stromal Cells: Expression Profiles of CYP2C19, FABP4, and FABP5 in TNBC Cells Co-Cultured with Stroma Cells 

To investigate whether cells present in the tumor microenvironment can influence metastatic TNBC cell behavior and expression of CYP2C19, FABP4, and FABP5, we used an in vitro co-culture model, selecting some of the representative tumor stromal cells, namely fibroblasts, adipocytes, and cells of monocytic lineage known to influence lipid homeostasis in breast cancer cells [8,43,44,45,46]. Prior to co-culture with TNBC cells, first, we examined the adipogenic markers in undifferentiated fibroblasts (3T3-L1), a stable adipocyte differentiated cell line derived from 3T3-L1 (3T3-adipocytes); an undifferentiated monocytic cell line (THP-1), peroxisome proliferator-activated receptor-γ (PPAR-γ), which regulates adipocyte differentiation; CCAAT/enhancer binding protein α (C/EBPα) which is responsible for mitotic growth arrest and late-stage adipocyte differentiation; and FABP4, FABP5, and CYP2C19 in the THP-1 and 3T3-L1 cells. The effect of pan-PPAR agonist rosiglitazone (2 µM), used to differentiate 3T3-L1 cells to mature adipocytes [47], was examined parallel with THP-1 cells. Oil red O staining showed accumulation of lipid droplets in adipocytic cells, differentiating them from 3T3 fibroblasts. Western blotting data show that rosiglitazone induced significant expression of FABP4, FABP5, and PPAR-γ in THP-1 cells, while differentiated 3T3-adipocytes had the highest expression of FABP4, FABP5, and PPAR-γ among the tested cells. Representative Western blots from three independent experiments are shown in Figure 4a. Corresponding statistical and densitometry ratio analyses are presented in Appendix A.

We further checked the expression levels of CYP2C19, FABP4, and FABP5 in 231-iR2L and LM6-derived knockdown cells co-cultured with 3T3 fibroblast, 3T3-derived adipocytes, or THP-1 cells. Representative Western blots from three independent experiments are shown in Figure 4b. Corresponding statistical and densitometry ratio analyses are presented in Appendix A. We observed an increase in FABP4 and FABP5 protein levels in LM6-shFABP4 and LM6-shFABP5 cells co-cultured with adipocytes as compared with cells grown in monoculture. Co-cultured with fibroblasts or 3T3-adipocytes, FABP4 protein level was increased in shCYP2C19 cells as compared with monoculture LM6-LacZ, LM6-shFABP4, or LM6-shFABP5 cells. Furthermore, upon co-culture with THP-1, FABP5 expression level was increased in FABP4- and FABP5-depleted cells. Expression of CYP2C19 was highest in monoculture of LM6-shLacZ cells and lowest in LM6-shLacZ cells co-cultured with fibroblasts or THP-1 cells. There was a slight increase in CYP2C19 levels of LM6-shFABP5 cells co-cultured with adipocytes. Gene expression levels for FABP4, FABP5, and CYP2C19 in the TNBC cell lines were not altered upon co-culture as compared with cells grown in monoculture (Appendix A). These results indicate that co-culture with adipocytes or monocytes influences the intrinsic expression of CYP2C19, FABP4, and FABP5 proteins in TNBC cells that support in part the interaction of TNBC tumor cells and stroma cells in TME. 

### 2.5. Potential Mechanistic Involvement of Stromal Cells: Co-Culture with Adipocytes or Monocytes Induces Metastatic Phenotype and Upregulation of FABP-EET Network in TNBC Cells

We compared the metastatic phenotypes of TNBC cells upon transwell co-culture with fibroblasts, adipocytes, or monocytes. Colony formation of LM6-shLacZ control cells grown in monoculture or after 10 days co-culture with any of the three cell types was significantly higher as compared with the 231-iR2L or FABP-/CYP-knockdown clones (Figure 4c). Interestingly, co-culture with adipocytes induced colony formation in LM6-shFABP4 cells, and co-culture with THP-1 significantly increased colony formation of LM6-shFABP5 and LM6-shCYP2C19 cells. The migration capacity of 231-iR2L, LM6-shLacZ, and LM6-shFABP4 cells were remarkably increased upon co-culture with adipocytes, but the values for LM6-shLacZ, and LM6-shFABP4 cells were greater than 231-iR2L cells (Figure 4d). The migration of the LM6-shFABP5 cells was enhanced after co-culture with THP-1 (*p* < 0.05). Similarly, adipocyte co-culture increased invasion potential of the LM6-shLacZ and FABP4-depleted cells while co-culture with THP-1 induced a more invasive phenotype in shFABP5 cells (Figure 4e). The invasion and migration potential of CYP2C19-depleted cells was not affected by co-culture with any of the three stromal cells. Interestingly, no significant difference was observed for the colony formation, migration, and invasion capacity of all cells co-cultured with fibroblasts as compared with monocultured controls. These data suggest that adipocytes (3T3-L1 differentiated adipocytes) or monocytes (THP-1) in the tumor microenvironment promote migration and invasion in parental (231-iR2L) and lung-seeking (LM6) TNBC cells that are potentially through paracrine factors or compensation of the FABP4 or FABP5 functions. 

In addition, we measured the changes in the expression of selected key proteins in the FABP-EET signaling network, identified from in silico analysis, following 24 h co-culture experiments. Representative Western blots from three independent experiments are shown in Figure 4f. Corresponding statistical and densitometry ratio analyses are presented in Appendix A. Protein levels of FABP-EET network proteins involved in stromal cell interactions (ezrin, CD44, and MMP9), metastasis (Src and Fak), EMT (vimentin), stem cell transformation (c-myc and Sox-2), and fatty acid-related proliferation protein (CD36) were intrinsically upregulated in LM6-LacZ as compared with the parental 231-iR2L cells. FABP4, FABP5, or CYP2C19-knockdown resulted in a reduction of protein expression of these markers. This pattern was conserved following co-culture with fibroblasts. Co-culture with adipocytes, however, resulted in a significantly increased expression of CD44, MMP-9, vimentin, and CD36 in LM6-shFABP4 cells. When co-cultured with THP-1 cells, Sox-2, vimentin, p-Src^419^, MMP9, and Ezrin protein levels in FABP5- and CYP2C19-depleted cells were upregulated as compared with the respective monoculture cells. These results suggest that stromal or microenvironment factors influence the activation of metastatic cascades facilitated by FABP/EET/CYP signaling.

### 2.6. In Vivo Functional Analysis: CYP2C19, FABP4, and FABP5 Differentially Modulate Primary LM6 Tumor Growth, Relapse, and Metastasis In Vivo

To further investigate the roles of the CYP2C19, FABP4, and FABP5 in TNBC progression, we established an orthotopic xenograft murine model using the 231-iR2L, LM6-shLacZ, LM6-shCYP2C19, LM6-shFABP4, and LM6-shFABP5 cells. The rate of primary tumor formation, lung metastasis, and local relapse after tumor resection was monitored. The organ indices, EET levels in lung and tumor tissues, and expression levels and co-localization of selected protein markers were measured at day 28 post-tumor resection. Surprisingly, before resection, the local tumor growth rate was significantly slower in the shFABP5 and shCYP2C19 groups (Figure 5a,b). Resection time points were set out at day 24 for the 231-iR2L, LM6-shLacZ, and LM6-shFABP4 groups, and at day 42 for the the LM-shFABP5 and LM6-shCYP2C19 groups, when the tumor volumes reached approximately 400 to 500 mm^3^. Whole body fluorescence was taken to monitor local tumor relapse and lung metastasis (Figure 5c). Representative whole animal body and lung organ fluorescence images are presented in Figure 5d. Post-resection formation of lung metastasis was higher in LM6-shLacZ-bearing mice than that of the 231-iR2L mice, and few metastatic nodules were observed in LM6-shFABP5 mice, but not in LM6shFABP4 and LM6-CYP2C19 mice (Figure 5d, inset). By day 28 post-tumor resection, 36% (5/14) LM6-LacZ, 22% (3/14), and 14% (2/14) of the LM6-sHFBP5 animals developed lung metastasis (Figure 5e). Conversely, local tumor relapse developed in all the LM6-shLacZ animals (14/14 mice), 70% (10/14 mice) of the 231-iR2L group, and 50% (7/14 mice) of the LM6-shFABP4 group; remarkably, relapse was much lower at 20% (3/14 mice) in LM-shFABP5 and LM6-shCYP2C19 groups (Figure 5f). 

We then examined the levels of Ki67, FABP4, FABP5, and CYP2C19 in tumor and lung tissues by IHC staining (Figure 5g). IHC Profiler [48] was used to quantify the percentile distribution in the tissues for strong positive (≥33%), positive (≥33%), and weak positive/negative staining (≥33%) cells (summation of the three categories of cell are ≅100%) (Figure 5h). The LM6-shLacZ tumors had strong Ki67 scores (47% strong positive and 43% positive), significantly higher than tumors from 231-iR2L (52% positive), LM6-shFABP4 (53% positive), LM6-shFABP5 (39% positive and 45% weak positive/negative), or LM6-shCYP2C19 (45% positive and 36% weak positive/negative) groups. Interestingly, positive FABP4 staining was observed for 231-iR2L (53% positive), LM6-shLacZ (49% positive), and even LM6-shFABP4 (44% positive) tumors; the other two groups LM6-shFABP5 and LM6-shCYP2C19 were classified as positive or weak positive/negative. Meanwhile, higher FABP5 staining/scores were observed for the LM6-shLacZ (34% strong positive and 43% positive) and LM6-shFABP4 (49% positive) tumors than that of the 231-iR2L (40% positive and 46% weak positive/negative), LM6-shFABP5 (39% positive and 52% weak positive/negative), and LM6-CYP2C19 (33% positive and 56% weak positive/negative). The CYP2C19 expression for 231-iR2L (56% positive and 33% weak positive/negative), LM6-shLacZ (61% positive), LM6-shFABP4 (62% positive) and LM6-FABP5 (65% positive) were all classified as positive while only the LM6-CYP2C19 (37% positive and 57% weak positive/negative) tumors were classified as weak positive/negative. Together, LM6-shLacZ control tumors, aside from having a more metastatic phenotype, were also highly proliferative and had high expression levels of CYP2C19, FABP4, and FABP5; surprisingly, shFABP4, shFABP5, and shCYP2C19 tumors were also detected with positive FABP4 (44%), FABP5 (39%), and CYP2C19 (37%) staining, respectively. 

In addition to the local tumors, we also examined the expression of the four proteins in the lungs from the same batch of animals. Tumor nodules with positive Ki-67 staining were detected in the lungs of 231-iR2L (36% positive and 49% weak positive/negative), LM6-shLacZ (48% positive), and LM6-shFABP5 (37% positive and 44% weak positive/negative) groups, and little or marginal expression was found in shFABP4 (83% weak positive/negative) and shCYP2C19 (81% weak positive/negative) groups. Expressions of FABP4 in the lungs of all the animal groups were as 39% positive and 49% weak positive/negative for LM6-shLacZ, 42% positive and 52% weak positive/positive for LM6-shCYP2C19, and 231-iR2L and LM6-shFABP5 were weak positive/negative. Strong positive expression of FABP5 in the lungs of LM6-shLacZ (43% strong positive and 35% positive) and weak positive/negative in the lungs of 231-iR2L (65%), LM6-shFABP4 (57%), and LM6-shCYP2C19 (52%) were observed. FABP4 and FABP5 expression levels were still detected 40% positive and 24% strong positive and 63% positive in the lungs of LM6-shFABP4 and shFABP5, respectively, while CYP2C19 level was detected with 65% to 80% weak positive/negative for all the groups. In summary, the IHC data from tumors and lungs suggest that the expression of FABP4, FABP5, or CYP2C19 is not solely come from cancer cells but also from resident or infiltrating stromal cells in tumor and lung microenvironments, affecting TNBC tumor growth or metastasis.

We further examined the EET concentrations in the resected tumors and whole lung tissues from the same mice. The complete sets of quantified oxylipin concentrations in the TNBC tumors and corresponding lung tissues are presented in Appendix A. PLS-DA clustering and oxylipin profile of the tumor and lung tissues are presented in Figure 5i. Corresponding scores and loading plots show that metabolites mainly derived from CYP epoxygenases and hydroxylases, including the four EET isomers, are responsible for the distinct clustering of the LM6 tumors and lung tissues, indicative of their crucial role in the metastasis of this lung-seeking TNBC cell clone. Congruent with the IHC results showing the specific protein expressions in the tumors and lungs, and the in vitro co-culture experiments with stromal cells, this clustering analysis shows that CYP2C19, FABP4, and FABP5 expression and EET levels are influenced by stromal cells at the tumor and metastatic sites; furthermore, depletion of both lipid chaperones in TNBC cells are crucial to inhibit FABP-EET-mediated relapse or metastasis.

### 2.7. Therapeutic Implications: FABP4 and FABP5 Inhibition Attenuates TNBC Metastasis In Vitro 

On the basis of our in vivo and in vitro co-culture and EET-supplementation experiments, we hypothesized that inhibiting FABP4 and FABP5 is critical in attenuating EET-driven TNBC tumor growth, relapse, and metastasis. To explore the potential of targeting these axes as a therapeutic approach for metastatic TNBC, we used a specific inhibitor of FABP4 activity (BMS309403), a pan-PPAR antagonist and FABP5 ligand (GW0742), and all-trans retinoic acid (ATRA), a compound known to activate PPAR signaling via FABP5. We also determined the effects of 15-deoxy12,14-PGJ_2_ (15-d-PGJ_2_), an endogenous ligand of PPAR-γ with anti-proliferative properties independent to FABP4 and FABP5 signaling [45]. The addition of BMS309403 (100 nM), GW0742 (100 nM), or 15-d-PGJ_2_ (100 nM) alone, did not have any significant effect on the migration (Figure 6a) and invasion (Figure 6b) potential of LM6 cells grown in monoculture, with or without 10 nM EETs supplementation. Treatment with ATRA (75 nM), induced an invasive phenotype with or without exogenous EETs. Notably, combination treatment with GW0742 (100 nM) and BMS309403 (100 nM), significantly decreased the migration and invasion capacities of LM6 cells. (Figure 6a,b). We have observed a minimal effect on LM6 cell viability (80–100%) following 24 h treatment with 75 nM ATRA, or 100 nM 15-d-PGJ_2_, 100 nM BMS309403 and 100 nM GW0742 alone or in combination, ruling out the contribution of anti- or pro-proliferation activities of these compounds on TNBC cell migration and invasion (Figure 6c). Moreover, co-treatment with FABP4/FABP5 inhibitors effectively attenuated the migratory and invasive potential of LM6 cells co-cultured with stromal fibroblasts, adipocytes and monocytes (Figure 6d,e). In summary, these results demonstrate that development of agents that have the capacity to regulate EET/FABP4/FABP5-mediated metastatic signaling simultaneously is an effective approach against TNBC relapse and metastasis.

### 2.8. Therapeutic Implications: dLGG-Treatment Suppresses TNBC Relapse and Metastasis by Regulating the FABP/EET-Mediated Signaling In Vivo

To validate whether targeting the FABP-EET axis can indeed attenuate TNBC metastasis, we used a plant-derived bioactive galactolipid dLGG, which has been shown to deregulate the oxylipin dynamics in murine models of sepsis and melanoma [31,33]. Doxorubicin, a chemotherapeutic drug for TNBC, was used as a positive control in this study. We observed decreased expressions of FABP4, FABP5, PPAR-γ, and SREBP-2 in LM6 cells treated with 40 µM dLGG independent of its anti-proliferative activity (>80% viable cells detected). Representative Western blots from three independent experiments are shown in Figure 7a. Corresponding statistical and densitometry ratio analyses are presented in Appendix A. dLGG (40 µM) likewise inhibited the migration and invasion potential of LM6 cells grown in monoculture or co-culture (Figure 7b,c). To investigate the efficacy of dLGG (25 mg/kg daily, o.p., dLGG25) against TNBC primary tumor growth, local relapse, and metastasis, we used a tumor-resection orthotopic xenograft model with LM6-iR2L cells. Doxorubicin (5 mg/kg every 3 days, i.p., DOX5) alone or in combination with dLGG25 were designed to assess the potential of dLGG in reducing doxorubicin-induced side effects. Compared with the LM6-iR2L tumor control group, the rate of original tumor or relapsed growth for mice with mono or combinational treatments were significantly slower (*p* < 0.5) (Figure 7d,e). Animals treated with DOX5 lost a significant amount of weight, which was not seen in the dLGG25 or DOX5 + dLGG25 groups, suggesting that dLGG treatment reverses this side effect of doxorubicin treatment, although no synergism was observed for the antitumor effect in the combination treatment (Figure 7f). Organ indices showed that the observed increase in lung and spleen weight in the LM6-inoculated mice was reversed by dLGG (Figure 7g). Moreover, the median survival time of the tested mice were 69 days, 60 days, 48 days, and 42 days observed in the dLGG25, dLGG25 + DOX5, DOX5, and control mice, respectively. The overall life span of the dLGG25 + DOX5 animals was 75 days, which was longer than DOX monotherapy (52 days) and could be due to the alleviation of doxorubicin-induced body weight loss by dLGG (Figure 7h). These data indicate the novel efficacy of dLGG on prolonging animal life. 

We also examined the expression levels of Ki67, FABP4, and FABP5 in tumors, at the resection time point, which were significantly attenuated by treatment with dLGG and doxorubicin, alone or in combination (Figure 7i). On the one hand, quantification results show that the LM6 tumor control had strong positive (>50%) scores for the three proteins. The dLGG-treated mice, on the other hand, had low positive scores (~50% positive), while doxorubicin and dLGG+DOX combination treatment had low positive to negative scores (~80% low positive/negative) for Ki67, FABP4/5 (Figure 7i). These findings demonstrate that dLGG and doxorubicin inhibit primary tumor growth, relapse, and metastasis at least in part via dual deregulation of FABP4 and FABP5 in metastatic TNBC tumors. 

We also examined the overall change in oxylipin dynamics in TNBC tumor and lung tissues of LM6 control and dLGG- or DOX-treated mice (Appendix A). The levels of all the EET isomers were lower in the three treatment groups as compared with the LM6 tumor control group (Figure 7j). Oxylipin dynamics in the tumors and lungs of LM6-inoculated mice were also perturbed by dLGG and DOX treatment (Appendix A). These findings show, for the first time, that doxorubicin treatment modulates the oxylipin profile of murine TNBC tumors. We further highlight that dLGG attenuates TNBC aggressiveness by regulation of FABP-EET signaling axis and could have the potential for future drug development for metastatic TNBC patients. 

## 3. Discussion 

To date, targeted approaches for TNBC therapy remains elusive, and a better understanding of the molecular intricacies of this aggressive BC subtype may aid in uncovering effective strategies for better intervention outcomes [49]. Moreover, the interplay between endogenous lipid metabolite levels and signaling by FABPs has not been explored in TNBC oncogenic progression. This study unveils novel roles for CYP2C19, FABP4, and FABP5 in TNBC relapse and metastasis. Using public datasets, we showed that high expression levels of these three proteins correlated with TNBC patient survival and metastasis profile. Results from our most recent study [25] also support the rationale and the potential clinical relevance of the current findings. We have shown, using fresh tumor specimens that EETs and CYP2C19 were upregulated in TNBC tumors, which correspond to the co-upregulation of fatty acid binding proteins, adipocyte signaling, and metastasis-related processes deciphered using pathway deregulation analysis [25]. Furthermore, increased expression of FABP4 and FABP5 in highly metastatic TNBC cells (LM6) as compared with parental MDA-MB-231 facilitated CYP2C19-derived EET-driven proliferation and metastasis transformation through the activation of oncogenic cascades involved in proliferation, EMT, and stemness-like formation. In vitro and in vivo results showed that in lung-seeking TNBC cells, this signaling axis is functionally associated with EET-mediated metastatic phenotype and stromal cell interactions. These results recapitulate our previous findings that EET metabolites mediate metastasis transformation in the mesenchymal-like TNBC cell line MDA-MB-231. Future studies are warranted to identify other unique features of this TNBC cell subtype, and to elucidate the biological significance of the FABP/EET network in heterogenous tumor samples.

Although FAPB5 has previously been linked with BC proliferation and FABP4 with lymph node metastasis [14,21,22,23,24], doubling times and proliferation were not altered in the LM6 knockdown clones, suggesting that a complex regulation mechanism could be in place, especially in highly metastatic mesenchymal-like TNBC cells. Depletion of CYP2C19, FABP4, and FABP5 resulted in the decreased expression of metastasis-related proteins and a general decrease in EETs, which subsequently suppressed migration, invasion, and colony formation potential of lung-seeking TNBC cells. In vitro EET supplementation and direct fibroblast/adipocyte/monocyte co-culture experiments further showed that TNBC cells uptake EETs from exogenous sources which is partly compromised when FABP4 or FABP5 is depleted, suggesting their critical role in facilitating EET-associated intracellular signaling. We consider that a mutual downregulation of CYP2C19 and FABP4/5 protein expression could be through a possible feedback mechanism induced at the metabolite level by the intrinsic EETs present in TNBC cells. Several routes of potential mechanisms by which metabolites regulated protein function, cell signaling, and gene expression have been put forward [50,51]. These routes include involvement of functional regions in the protein/RNA, metabolite sensing via the sensor-transducer-effector model, and highly specific metabolite–macromolecule interactions. Possibly, the presence of intrinsic EETs affect the expression of CYP2C19/FABP4/FABP5 through the mechanisms exemplified in these recent works. Direct protein–protein or metabolite–protein interactions of CYP2C19/FABP4/5 and EETs, however, have not been identified. Future directions stemming from the current work are geared towards elucidating these interactions. Moreover, the addition of exogenous EETs in the culture media induced the nuclear translocation of FABP4 and FABP5, and nuclear accumulation of PPAR-γ and SREBP-2 in lung-seeking TNBC cells. These results indicate that FABP/EET-mediated activation of oncogenic TFs cascades facilitate the proliferative and metastatic network in TNBC, inhibition of which is a viable target to attenuate tumor growth and metastatic transformation [52]. We demonstrated, both in vitro and in vivo, that FABP4 and FABP5, and their associated CYP2C19/EET signaling axis provide a regulatory mechanism in TNBC metastatic progression, regardless of whether they are intrinsically expressed by cancer cells or are activated through interaction with other cell types. Enrichment of FABP/EET-source cells, for example, adipocytes and monocytes, in the metastasis niche, can be one method by which distal homing of FABP5-deprived cancer cells occurs, even in the absence of the primary tumor (tumor resection model). Although further mechanistic studies are needed to understand the dynamics of FABP4 and FABP5 acquisition and transcriptional activity in TNBC cells, or the influence of paracrine factors from stromal or other cells present in the local tumor or metastasis sites, for example, cytokines, adipokines, or endogenous lipophilic ligands, our work sheds light on the potential of targeting FABP4 and FABP5 activities in suppressing TNBC progression and metastatic transformation. 

Despite the toxic side effects and a limited therapeutic window, only chemotherapy currently holds promise for treatment and prolonged patient survival in metastatic TNBC. In this study, we demonstrated that the frontline chemotherapeutic drug for TNBC patients, doxorubicin, decreased the expression of FABP4 and FABP5, and perturbed the oxylipin dynamics in TNBC tumors and metastasis site. We also show that specific inhibitors of FABP4 and FABP5 and the plant-derived galactolipid, dLGG, inhibit the metastatic phenotype in TNBC cells and in tumor inoculated animals. Moreover, dLGG prolongs survival, attenuates doxorubicin-induced weight loss, and inhibits the aggressive behavior and metastasis of TNBC tumors in mice by concomitantly regulating lipid chaperones FABP4 and FABP5, and lipid mediator EET-related signaling networks. These results suggest that dual regulation of FABP4 and FABP5 could be effective against TNBC relapse and metastasis. 

## 4. Materials and Methods

### 4.1. Cell Lines and Culture Conditions 

Human TNBC (MDA-MB-231) and monocytic (THP-1) cell lines, and murine fibroblasts (3T3-L1) were from ATCC (Manassas, VA, USA). The MDA-MB-231 cells expressing luciferase and fluorescence dual reporter, iRFP-2A-Luc (231-iR2L), were constructed as previously described [48] (Appendix A). A lung-seeking 231-iR2L subclone (LM6) was derived from six serial rounds of spontaneous lung metastasis in SCID mice, via tail vein re-injection of primary lung-seeking cells, and then, selection by FACS (fluorescence activated cells sorter) (Appendix A). The 3T3-L1 fibroblasts were differentiated to adipocytes following published protocols [47]. Cells were cultured in RPMI-1640 (THP-1) and DMEM (3T3-L1 and MDA-MB-231) from ThermoFisher Scientific (Waltham, MA, USA) supplemented with 10% FBS, 100 units/mL penicillin, and 100 mg/mL streptomycin at 37 °C in a humidified 5% CO_2_ incubator. The culture conditions used were based on the recommendations of the National Cancer Institute Physical Sciences—Oncology Centers Network on the baseline ATCC protocol [53]. As recommended by ATCC for cell cultures grown in 5% CO_2_, 3.7 g/L sodium bicarbonate was added to the culture media adjusted to an initial pH of 7.6. Unsealed (loose caps) culture flasks or dishes were used to allow the gases to equilibrate. The shRNA-mediated depletion experiments were as described previously in [54]. The shRNA clones (Appendix A) were purchased from the National RNAi Core Facility Academia Sinica (Taiwan). 

### 4.2. Compound dLGG Preparation 

The dLGG was prepared from whole plant extract of *Crassocephalum rabens* (Asteraceae) as previously described in [32].

### 4.3. Orthotopic Xenograft Model and Establishment of the Metastatic Sublines

All experimental procedures were approved by the Institutional Animal Care and Utilization Committee of Academia Sinica, Taiwan (Protocol ID: 17-12-1171, approval date: 18 June 2018) and were performed in compliance with the Guide for the Care and Use of Laboratory Animals by the Ministry of Science and Technology and the Taiwan Animal Protection Law. Female NOD-SCID mice (5 weeks old) were obtained from the Laboratory Animal Core Facility (Agricultural Biotechnology Research Center, Academia Sinica, New Taipei, Taiwan). Tumor development was monitored by caliper measurements and IVIS imaging (Xenogen, Alameda, CA, USA). The functional roles of FABP4, FABP5, and CYP2C19, and the therapeutic effects of dLGG and doxorubicin against metastatic TNBC were evaluated using an orthotopic xenograft tumor model (231-iR2L, LM6, and FABP4-, FABP5- and CYP2C19-depleted LM6-derived subclones) (Appendix A). Whole body fluorescence quantification was performed every 7 days, 30 to 70 days post-tumor implantation. Schemes of animal experiments are shown in Appendix A.

### 4.4. Histology and Immunohistochemistry 

Tumor and organ tissues for H&E staining and immunohistochemistry (IHC) analysis were according to methods published elsewhere [25]. Images were captured using AxioVision (Carl Zeiss MicroImaging, White Plains, NY, USA). Automated quantification of tissue immunoreactivity was conducted using IHC Profiler; an Image J plugin [48]. Percentile score of negative/low positive, positive, and highly positive DAB-stained cytoplasmic zones were compared.

### 4.5. Western Blotting 

Mouse tissues (0.1 g) or cell pellets were extracted in 0.4 mL RIPA lysis buffer, homogenized in a mixer ball mill (MM301, Retsch, Haan, Germany) for 2 min, and centrifuged at 15,000× *g* for 30 min at 4 °C. Nucleocytoplasmic fractionation was performed by scraping cells (with or without treatment) from culture dishes and washing twice with cold PBS. Cells were resuspended in hypotonic buffer (20 mM Tris-HCl, pH 7.4, 10 mM NaCl, 3 mM MgCl_2_), incubated on ice for 15 min, gently homogenized, and then, centrifuged for 10 min at 3000 rpm at 4 °C. Supernatants containing the cytoplasmic fraction were transferred and saved for SDS-PAGE. Pellets containing the nuclear fraction were resuspended in lysis buffer and centrifuged for 30 min at 15,000 rpm at 4 °C. Resulting supernatants were transferred to clean microtubes and quantified. Protein concentrations for total cell homogenate, nuclear, and cytoplasmic fractions were determined using a DC protein assay kit (Bio-Rad). Protein was resolved by 12% SDS-PAGE and immunoblotted with specific antibody, and then, the reacted bands were visual by enhanced chemiluminescence reagents (ECL, Amersham). Antibodies used for Western blotting and immunohistochemical staining were the following: c-myc (ab32), Src (ab133283), and PPAR-δ (ab23673) from abcam (Cambrige, MA, USA); ezrin (3145), FAK (1688), p-FAK (3282S), Ki67 (9027S), PPAR-γ (2443), and p-Src (2105S) obtained from Cell Signaling Technology (Beverly, MA); CD44 (217594), FABP4 (15872-1-AP), FABP5 (12348-1-AP), Sox-2 (11064-1-AP), and vimentin (10366-1-AP) from Proteintech; actin (mab1501) from Millipore; and CD36 (sc-7309),C/EBPα (sc-9315), CYP2C9/19 (sc-23436), LXRα/β (sc-13068), MMP-9 (sc-6840), PPAR-α (sc-398394), PPAR-β (sc-74517), PXR (sc-25381), RhoA (sc-418), RARα (sc-515796), and RXRα (sc-515929) purchased from Santa Cruz (Santa Cruz, CA, USA). Additional antibody information for the western blot and immunohistochemistry analysis are presented in Appendix A. Western blotting experiments were performed using identically treated samples from three independent experiments/biological replicates. Band intensities for each protein of interest (POI) were determined by densitometry software (Image J). Housekeeping proteins (HKP) actin, GADPH, and lamin A/C were utilized as loading controls (LC) for whole cell lysates, cytoplasmic, and nuclear fractions, respectively. Normalized band intensity for each POI was calculated by multiplying the raw densitometry reading by the loading control normalization factor (ratio of HKP band intensity in the vehicle-treated and parental cell control lane to the HKP intensity in the target protein lane in each immunoblot). Relative protein expressions (as fold change relative to the corresponding treatment and cell control) were then calculated. Mean relative protein expressions and standard deviations were obtained from the three experiments and statistical analysis was performed using one-way ANOVA, post hoc Dunnett’s test, or unpaired Student’s *t*-test (*p* < 0.05), where applicable, by SPSS software (version 16.0). Quantified Western blotting information are presented as plots of relative protein expression in Appendix A. Corresponding densitometry intensity, normalization ratios, normalized protein intensities, fold changes relative to corresponding controls, and statistical analyses are presented in detail in Appendix A. Representative Western blot images with corresponding molecular weight markers for all protein bands presented in Figure 2b,c, Figure 3e,f, Figure 4a,b,f and Figure 7a are shown in Appendix A.

### 4.6. Real-Time Quantitative PCR

Total RNA was isolated using a RNeasy Mini Kit (Quiagen, Germantown, MD, USA) and respective cDNA was generated by reverse transcription of RNA aliquots using the Takara PrimeScript RT Reagent Kit (Takara, Mountain View, CA, USA) according to the manufacturers’ instructions. The resulting cDNA was used for real-time PCR with SYBR Premix Ex Taq Kit (Takara) in a StepOne Real-Time PCR Detection System (Life Technologies, Camarillo, CA, USA). Primer information are presented in Appendix A. Expression data were normalized to GADPH. 

### 4.7. Migration, Invasion, Colony Formation, and Cell Mobility Assays

Cell proliferation, migration, invasion, and colony formation were performed as previously described in [25]. Transwell (6.5 mm diameter, 8 mm pore size; Costar, Cambridge, MA, USA) migration experiments were as follows: Cells were placed into the upper chamber (5 × 10^4^ cells per insert) and DMEM culture medium containing 0.1% FBS. FBS (10%) was added to the lower well. Nonmigrating cells were scraped from the upper membrane surface after 24 h incubation. For invasion experiments, 8 mm filters were precoated with Matrigel (30 mg/filter) prior to following the migration assay protocol. The anchorage-independent colony formation assay was performed by growing cells in 24-well plates (500 cells/well) for 10 days. Migration, invasion, and colony formation were quantified by visualization at endpoint with 0.1% (*w/v*) crystal violet in PBS and quantitatively measuring absorbance at 590 nm. Cell motility was measured by recording cell trajectories for 18 h using an inverted Zeiss Axiovert 200 M microscope with an environmental chamber and phase-contrast optics (images taken every 30 min). Metamorph software (Molecular Devices) was used to evaluate migration velocities.

### 4.8. Oxylipin Metabolome Analysis 

Levels of bioactive lipid mediators in quiescent cells and frozen mouse tumor and lung tissue samples were determined using optimized in-house ultra-performance liquid chromatography-mass spectrometry (UPLC-MS/MS) [25,31].

### 4.9. Statistical Analysis 

Quantified data were expressed as mean ± SD. Statistical comparisons between sample groups and histoclinical parameters were calculated using unpaired Student’s *t*-test or ANOVA with post hoc Tukey and Dunnett’s test, where applicable, by SAS software (version 9.4) or SPSS (version 16.0). Multivariate partial least square-discriminant analysis (PLS-DA) was carried out using SIMCA-P 14.0 software (Umetrics, Umea, Sweden). The robustness of the classifier models was validated in a built-in iterative seven-fold leave-one-out approach. Oxylipin metabolites consistently detected in at least 80% of samples were included in the analyses. All known artifact peaks were excluded.

## 5. Conclusions

This study describes a novel mechanism by which mesenchymal-like TNBC cells utilize intrinsic or stroma-derived EETs via the lipid chaperones FABP4 and FABP5 to drive metastasis and proliferation. In vitro, we demonstrate that FABP4, FABP5, and EET induce the nuclear accumulation of PPAR-γ and SREBP-2-associated lipogenic and pro-metastatic signaling pathways. We further demonstrate the in vivo efficacy of phytogalactolipid dLGG and doxorubicin against FABP/EET-mediated TNBC tumor relapse and metastasis. Clinical implications of the findings from this study may further be strengthened by further validation experiments using patient-derived tumor samples and information. Comparative studies using a larger dataset is warranted to assess and strengthen the prognostic and translational applications of targeting the FABP/EET signaling axis as an approach to combat metastatic TNBC.

## Figures and Tables

**Figure 1 cancers-12-00199-f001:**
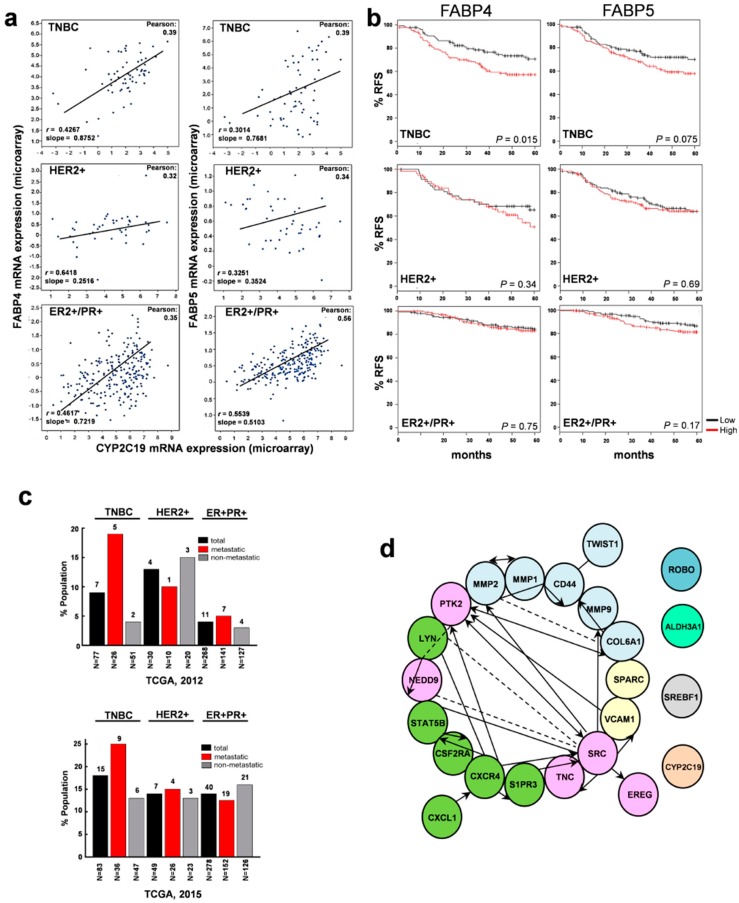
Concerted upregulation of CYP2C19, FABP4, and FABP5 is associated with metastasis transformation, stromal interaction network, and poor prognosis in TNBC. (**a**) FABP4 and FABP5 mRNA expression shows a positive co-expression profile when plotted against CYP2C19 epoxygenase in breast cancer tissues (two-tailed Pearson’s correlation analysis, *p* < 0.05, dataset from [25]). (**b**) Kaplan–Meier plots show relapse-free survival rates (RFS) of breast cancer patients classified according to hormone receptor subtype and stratified by either FABP4 or FABP5 mRNA expression level in tumors [34]. (**c**) Population distribution (%) of patients with concurrent CYP2C19, FABP4, and FABP5 upregulation in two independent, non-overlapping TCGA cohorts [25]. (**d**) Top upregulated genes in the FABP/CYP epoxygenase network visualized by cytoscape with a cut off value of significant relationships that was set by the Benjamini–Hochberg procedure (*FDR* < 0.01). Colors denote unique associated genes/pathways and arrow direction shows a canonical upstream/downstream relationship. Dashed lines present indirect interactions and solid lines denote direct interactions.

**Figure 2 cancers-12-00199-f002:**
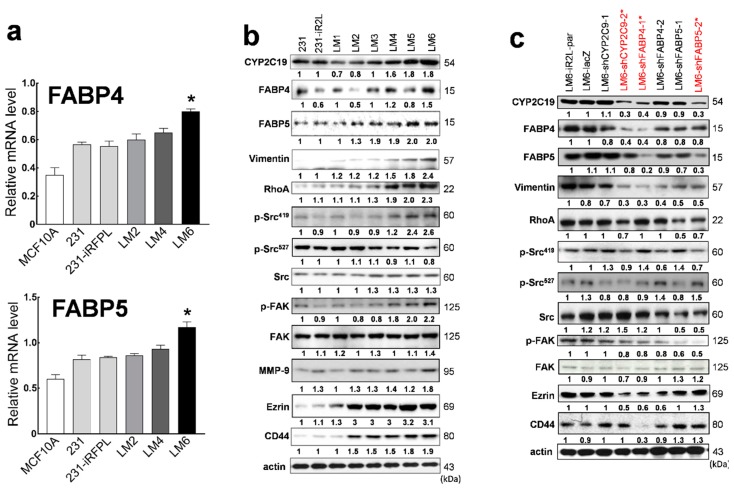
Lung-seeking and highly metastatic MDA-MB-231 TNBC cells are characterized by increased FABP4 and FABP5 gene and protein expressions and elevated EET levels. (**a**) Gene expression of FABP4 and FABP4 are significantly upregulated in LM6 cells as compared with immortalized mammary epithelial cells (MCF10A), parental 231, or 231-iR2L, and earlier metastatic subclones LM2 and LM4; (**b**) immunoblot analysis shows increased expression of FABP4, FABP5, and CYP2C19, as well as metastasis, EMT, and stromal interaction-related markers in acclimated lung-seeking subclone of MDA-MB-231 cells (LM6), which were decreased in the specific gene knockdown cell clones, LM6-shFABP4, LM6-shFABP5, and LM6-shCYP2C19; (**c**) representative blots from three independent experiments are shown. shRNA clones with asterisks were used in subsequent experiments; (**d**) box plots show the basal intracellular concentration of AA-derived EET isomers (5,6-EET, 8,9-EET, 11,12-EET, and 14,15-EET) in the parental MBA-MB-231 and 231-iR2L TNBC cells, its lung-seeking LM6 subclone, and in FABP4, FABP5, or CYP2C19-depleted LM6 cells analyzed using UPLC-MS/MS spectrometry; and (**e**) corresponding intracellular EET levels of each the cell lines under study were compared following 24 h culture in media supplemented with 10 nM of a specific EET isomer. All analyses include 3 biological replicates and 4 technical replicates. Error bars indicate mean ± SEM. Statistical significance of the data between different groups are denoted by different letters or asterisks, when applicable (*p* = 0.05, ANOVA, post hoc Tukey).

**Figure 3 cancers-12-00199-f003:**
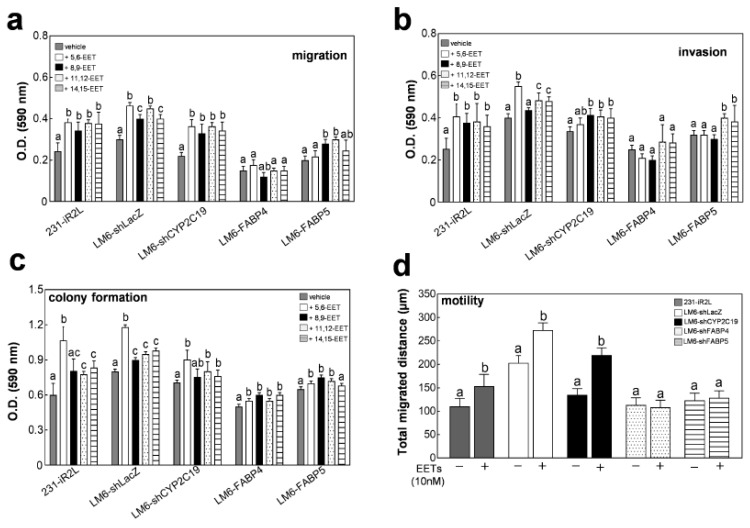
Nuclear translocation of FABP4 and FABP5 drives EET-mediated aggressiveness in lung-seeking metastatic MDA-MB-231 TNBC cells: (**a**) Migration; (**b**) invasion; (**c**) colony formation, and (**d**) motility of each the cell line under study were compared following 24 h culture in media supplemented with 10 nM of a specific EET isomer; (**e**) cellular localization of FABP4 and FABP5, and expression of lipogenic transcription factors PPAR-γ, SREBP-2, RAR, and RXR-α following supplementation of EETs to LM6 cells was determined by Western blotting (WCL, whole cell lysate; N, nuclear fraction; and C, cytosolic fraction); and (**f**) expression of downstream direct targets of PPAR-γ and SREBP-2 in LM6-LacZ, LM6-shFABP4, and LM6-shFABP5 cells grown with or without EET supplementation (10 nM, 2.5 nM each isoform) was determined by Western blotting. All analyses include 3 biological replicates and 4 technical replicates. Error bars indicate mean ± SEM. Statistical significance of the data between different groups are denoted by different letters (*p* = 0.05, ANOVA, post hoc Tukey).

**Figure 4 cancers-12-00199-f004:**
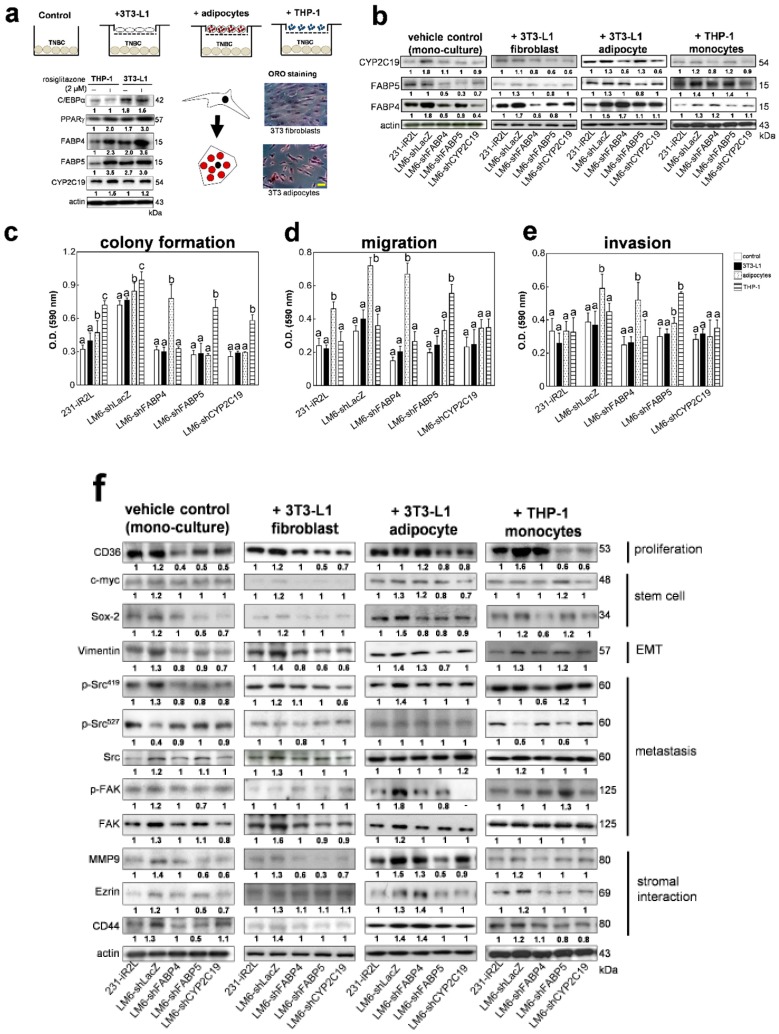
Expression of FABP4 and FABP5 are induced by co-culture of metastatic TNBC cells with stromal cells. (**a**) Schematic diagram showing co-culture of TNBC cells (1 × 10^6^) with 3T3-L1 murine fibroblasts, 3T3-L1 differentiated adipocytes, and monocytes THP-1. (Left) Western blot analysis of adipocyte markers and intrinsic expression levels of FABP4, FABP5, and CYP2C19 in fibroblasts, adipocytes, and monocytes grown in monoculture, with or without the addition of 2 µM rosiglitazone (adipocyte differentiation factor and PPAR-γ agonist). (Right) Oil Red O staining differentiates 3T3 and mature adipocytes, following differentiation at 10 days; scale bar for images set at 10 μm (**b**) expression levels of FABP4, FABP5, and CYP2C19 in 231-iR2L, LM6, and FABP4/FABP5CYP2C19-depleted cells co-cultured with stromal cells. The effects of co-culture on the (**c**) colony formation, (**d**) migration, and (**e**) invasion capacities of the TNBC cells; and (**f**) expression levels of FABP-EET network markers in MDA-MB-231, LM6, and FABP4/FABP5CYP2C19-depleted cells co-cultured with stromal cells. All analyses include 3 biological replicates and 4 technical replicates. Error bars indicate mean ± SEM. Statistical significance of the data between different groups are denoted by different letters (*p* = 0.05, ANOVA, post hoc Tukey).

**Figure 5 cancers-12-00199-f005:**
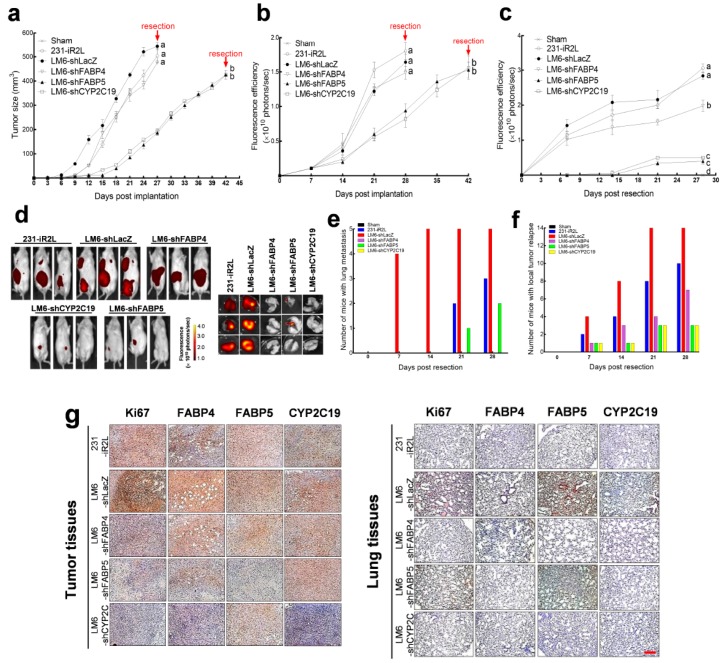
In vivo validation of the roles of CYP2C19, FABP4, and FABP5 in primary tumor growth, relapse, and metastasis. 231-iR2L, LM6, LM6-shFABP4, LM6-FABP5, and LM6-CYP2C19 were inoculated at 2 × 10^6^ cell density and the rate of tumor growth was measured (**a**) manually (caliper) or by (**b**) IVIS (whole body fluorescence imaging). (**c**) Whole body fluorescence was measured up to day 28 post resection and (**d**) representative images of animals and lung organs are presented for each group. (**e**) Number of test animals with lung metastasis lesions and (**f**) local tumor relapse are presented for each group. (**g**) IHC analysis and (**h**) corresponding quantification for the expression of Ki67, FABP4, FABP5, and CYP2C19 proteins in tumor and lung tissues. (**i**) Oxylipin profile and cross validated PLS-DA score and loading plots of tumor and lung tissues derived from all the tumor-bearing mice groups. Each biological replicate (*n* = 4) is represented by a single point. All analyses include 4 technical replicates per biological sample. Statistical significance of the data between different groups are denoted by different letters (*p* = 0.05, ANOVA, post hoc Tukey, scale bar for images set at 50 μm).

**Figure 6 cancers-12-00199-f006:**
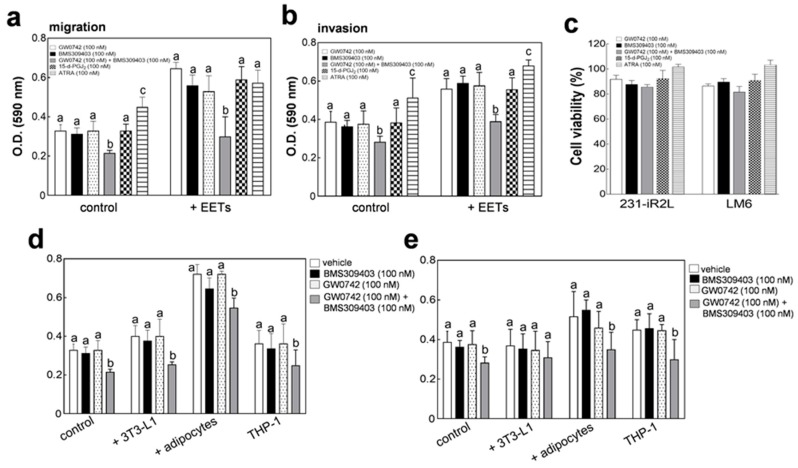
Dual regulation is necessary to abolish expression of FABP4 and FABP5 induced by EET-supplementation or co-culture of metastatic TNBC cells with stromal cells. Effects of a FABP4 inhibitor (BMS309403), a pan-PPAR antagonist (GW0742), an FABP5/PPAR-γ agonist all-trans retinoic acid (ATRA), and an endogenous PPAR-γ ligand, 15-deoxy12,14-PGJ_2_ (15-d-PGJ_2_) on the (**a**) migration and (**b**) invasion status of LM6 TNBC cells. (**c**) Viability of TNBC (231-iR2L and its lung-seeking subclone, LM6) cells at concentrations used for migration and invasion assays. Inhibitory effects of specific compound treatment on the (**d**) migration and (**e**) invasion potential of cells co-cultured with 3T3-L1 fibroblasts, adipocytes and THP-1 monocytes. All analyses include 3 biological replicates and 4 technical replicates. Error bars indicate mean ± SEM. Statistical significance of the data between different groups are denoted by different letters (*p* = 0.05, ANOVA, post hoc Tukey).

**Figure 7 cancers-12-00199-f007:**
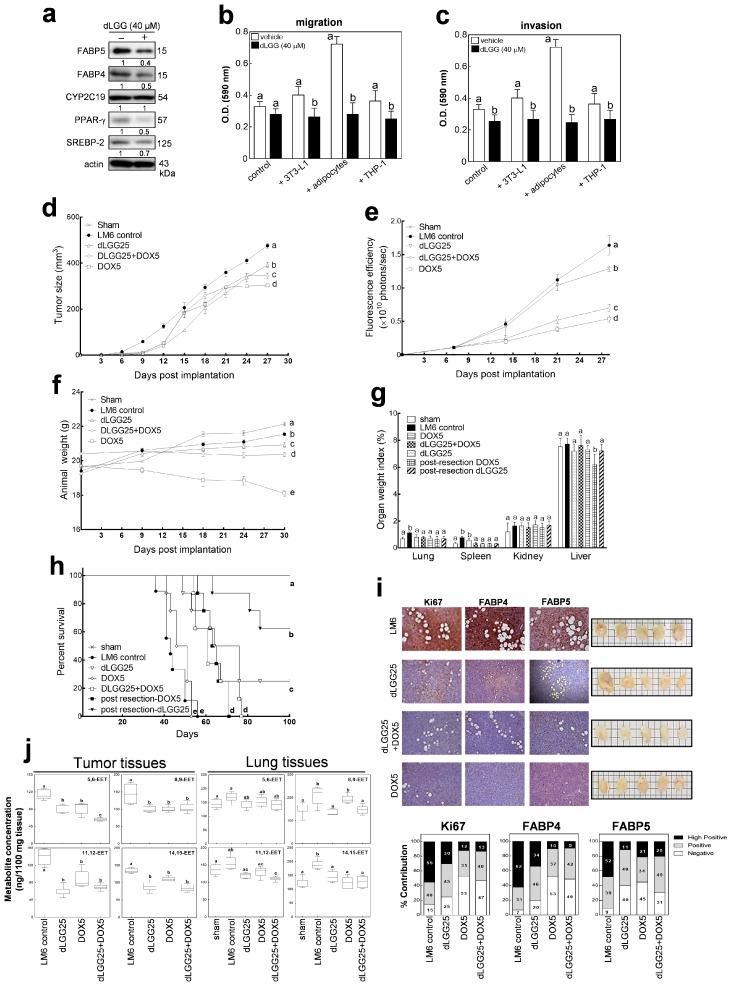
dLGG inhibits EET-FABP-mediated TNBC relapse and metastasis. The effects of dLGG-treatment (40 µM) on the expression level of (**a**) FABP-EET network proteins and (**b**) migration and (**c**) invasion status of LM6 TNBC cells. The NOD/SCID mice bearing lung-seeking metastatic TNBC tumors (LM6) were treated with vehicle, doxorubicin (DOX5 5 mg/kg daily i.p.), dLGG (dLGG25 25 mg/kg, daily o.p.), and a compound-drug combination of dox and dLGG (DOX5 + dLGG25). (**d**) Tumor size, (**e**) whole body fluorescence, and (**f**) animal and (**g**) tumor and tissue weight were monitored prior to tumor resection. (**h**) Tumors were resected when they reached 500 mm^3^ and Ki67, FABP4, and FABP5 expressions were analyzed by IHC and quantified using IHC profiler. (**i**) Survival was monitored up to 100 days post tumor inoculation. (**j**) Box plots showing the EET concentrations in the tumor and lung tissues of the different treatment groups at resection time point. Representative blots from three independent experiments are shown. Error bars indicate mean ± SEM, *n* = 14 mice per group. Significantly different values (*p* = 0.05, ANOVA, post hoc Tukey) are denoted by asterisks, scale bar for images, 50 μm.

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
