# Peer review of "Deregulating the CYP2C19/Epoxy-Eicosatrienoic Acid-Associated FABP4/FABP5 Signaling Network as a Therapeutic Approach for Metastatic Triple-Negative Breast Cancer"

_cancers, 2020, doi:10.3390/cancers12010199_

Round 1
Reviewer 1 Report
Manuscript and findings looks good. I don't have any comments at this point.
Thanks!
Author Response
We thank the Reviewer for the kind comments.
Reviewer 2 Report
The paper by Apaya et al. deals with the roles of FABP4, FABP5 and CYP2C19 in tumor growth and metastatic transformation of TNBC. In the course of their study, the authors actually analyze four different factors/agents (FABP4, FABP5, CYP2C19, EETs) in various cell lines/clones, cellular responses, markers, nuclear target transcription factors, readouts (primary tumor growth, relapse and metastasis, metastatic sides, impact of co-culture with different stromal cell types combined with different readouts in vitro and in vivo), and the effects of different inhibitors with different modes of action on TNBC responses in vitro (mono- and coculture) and in vivo, etc. They found that CYP2C19 and FABP4/5 are crucially involved in TNBC proliferation, metastatic progression and stromal interactions involving EET-induced nuclear translocation of FABP4/5, and activation of PPAR-gamma and SREBP-2. Moreover, they showed that the phytogalactolipd dLGG attenuated TNBC recurrence and and metastasis through regulation of FABP/EET.
Major points
While the manuscript is well written, it appears to be overloaded and lacks a clear focus. This is also reflected in Figure 7 which is far too complex. Sometimes the reader gets lost between the different factor, clones, readouts, regulatory interactions, targets and effects. Authors should streamline their manuscript and focus on selected issues and analyse these in-depth. I acknowledge the author’s intention to provide a comprehensive analysis of the three factors in vitro and in vivo but the wealth of data presented is at the cost of clarity. The authors should streamline the manuscript and focus either on the in vitro or the in vivo part. The in vitro part may focus on signaling, metastasis-relevant events such as EMT and EMT-associated responses like migration, invasion and stemness, while the in vivo part could focus on therapeutic aspects including evaluation of the natural compound, dLGG, alone or in combination with doxorubicin, which seems to be a quite promising agent/combination. The immunoblot data shown in Figures 2a, b, 3e, f, 5a, b, and f, and 6a need densitometry-based quantification and statistical calculation from three independent experiments. Quantification should be performed for those proteins for which quantitative measures were important with respect to the conclusions drawn. For instance lines 129-130: Vimentin, RhoA, CD44, ezrin, p-Src, p-FAK.Minor points
Lines 122+166: Designation incomplete: "sh” is missing in “LM6-FAPB5” and “LM6-CYP2C19”. The term “overexpression” was used a couple of times. I suppose authors do not refer to ectopic overexpression. However, if the claim is endogenous overexpression in their MDA-MB-based cellular model (not at the patient level), this would require a proper control, i.e. normal human breast epithelial cells or at least a benign immortalized breast cell line for quantitative comparison of FABPs, CYP2C19, and EET expression. MDA-MB-231 are per se metastatic. When comparing the parental cell line with the lung-seeking derivative, LM6, then it would be nice to test how FABP4/5 affect the interaction with lung cells and lung-specific proteins. For analysis of EMT, a better marker than RhoA would be E-cadherin Doubling times and proliferation were not altered in the knockdown clones despite the fact that FABP5 is linked to BC proliferation. Conversely, as shown in Figure 4a, shFABP4 cells do not show slowed tumor growth although previous studies have shown an association of this protein with aggressiveness and lymph node metastasis (Refs 14, 21-24). Is this a contradiction? The authors should comment on this. How can the mutual downregulation of CYP2C19 and FABP4/5 in the respective knockdown clones explained mechanistically (see Fig. 2b)? Lines 154-158: It is not clear to me why a differential decrease in EET isoforms upon FABP4 knockdown in LM6 cells is important. This should be explained in the manuscript. What are the letters “a” and “b” above columns in figures 3-6 supposed to mean? This must be clearly indicated in the figure legends. Line 357: The conclusion is incomplete. Adipocytes promote migration not only in lung-seeking TNBC but also in 231-iR2L cells (see lines 348-350). Lines 267-282: IHC results. It is not surprising that stromal cells also stain positive for the FABPs, CYP2C19, and EETs. In order to identify the tumor cell-specific expression in the tissue, the tumor cells should have been labeled either immunologically with a tumor marker or endogenously by transfection with GFP or similar.
Reviewer 3 Report
The authors demonstrated CYP2C19, FABP4 and FABP5 overexpression in relation to recurrence and metastasis of TNBC. A number of analyses have been performed including a xenograft mouse model, showing an approach to treat TNBC. The focus of this study is important in this field; however, I am afraid that the sample size of this study is highly limited to lead to draw definitive conclusions.
L20-21; ‘in TNBC tumors and in highly metastatic cell lines’>>> Experiments in this study is performed only in one TNBC cell line, MDA-MB-231. No fresh tumor sample was used together with the cell line. Because TNBC is heterogeneous, it is essential to test several samples and use different types of breast cancer samples as controls. A number of publications show comparative studies using various samples, for example:
Establishment of two basal-like breast cancer cell lines with extremely low tumorigenicity from Taiwanese premenopausal women.
Kuo WL, Ueng SH, Wu CH, Lee LY, Lee YS, Yu MC, Chen SC, Yu CC, Tsai CN.
Hum Cell. 2018 Apr;31(2):154-166.
There is a publication reporting a list of breast cancer cell lines.
Breast Cancer Cell Line Classification and Its Relevance with Breast Tumor Subtyping.
Dai X, Cheng H, Bai Z, Li J.
J Cancer. 2017 Sep 12;8(16):3131-3141.
I would like to authors to add results using additional TNBC cell lines to support conclusions.
Round 2
Reviewer 2 Report
The authors have satisfactorily responded to all my points of critique except for major point 2. Although they provided all the densitometric readings I wonder whether in fact these were derived from blots of independent experiments. What does the designation "sample 1/sample 2/ sample 3 mean? Are these identically treated samples from three independent experiments (starting with plating of cells, treatment, etc.)? I consider this unlikely because the values as well as the ratios with the corresponding values from the housekeeping genes (HKGs) are too similar. It appears to me that these are parallel samples (replicates) from ONE experiment. In order to calculate the means and SDs from three experiments, the ratios (GOI/HKG) have to be determined first from each of the three (or more) experiments. Only then the means and the SDs from the three experiments can be calculated and plotted.
The authors must explain step-by-step how they have carried out this procedure and more clearly label their tables so that the readers can see how the results of each of the three experiments look like.
Reviewer 3 Report
As authors explained, the results of this study are partly supported by a previous paper by Apaya et al. who showed a relationship between CYP2C19 and cancer migration, but FABP4 and FABP5 have not been examined in the previous study. Although data are not sufficient to lead the conclusion, the message in the title is overemphasized.
L581-582; Please describe the details of culture conditions for MDA-MB-231. As the cells are usually cultured without CO2, the results would be reflected by this different environment. Because expression profiles could be changed by culture conditions, the results need to be compared with a standard protocol suggested by ATCC.
Round 3
Reviewer 3 Report
In this study, the MDA-MB-231 cells were grown in the manufacturers' (ATCC) suggested medium, 5% CO2, 3.7 g/L sodium bicarbonate. In addiotion, unsealed (loose caps) culture flasks or dishes were used to allow the gases to equilibrate.
It is suggested by ATCC that MDA-MB-231 (ATCC® HTB-26™) is cultured using the Leibovitz's L-15 medium, which is not buffered by sodium bicarbonate. It is noted that 'the L-15 medium formulation was devised for use in a free gas exchange with atmospheric air and that a CO2 and air mixture is detrimental to cells when using this medium for cultivation.' Cells could be grown in a CO2 incubator using flasks when the caps must be tightly closed. Because the culture condition for the MDA-MB-231 cells in this study is irregular, affecting the expression profiles. Please refer to a previous paper by Ben-David et al., ‘Genetic and transcriptional evolution alters cancer cell line drug response.’ Nature. 2018;560:325-330
Additional experiments are required to support the results.
Author Response
Third Round Review Comments from Reviewer 3:
Comment 1: In this study, the MDA-MB-231 cells were grown in the manufacturers' (ATCC) suggested medium, 5% CO2, 3.7 g/L sodium bicarbonate. In addition, unsealed (loose caps) culture flasks or dishes were used to allow the gases to equilibrate.
It is suggested by ATCC that MDA-MB-231 (ATCC® HTB-26™) is cultured using the Leibovitz's L-15 medium, which is not buffered by sodium bicarbonate. It is noted that 'the L-15 medium formulation was devised for use in a free gas exchange with atmospheric air and that a CO2 and air mixture is detrimental to cells when using this medium for cultivation.' Cells could be grown in a CO2 incubator using flasks when the caps must be tightly closed. Because the culture condition for the MDA-MB-231 cells in this study is irregular, affecting the expression profiles. Please refer to a previous paper by Ben-David et al., ‘Genetic and transcriptional evolution alters cancer cell line drug response.’ Nature. 2018;560:325-330
Additional experiments are required to support the results.
Author's Response to Comment 1:
We acknowledge the Reviewer's comment and concern regarding the culture condition of MDA-MB-231 cells and its potential effect on the expression profiles. The culture conditions we used are/were based on the updated cell culture recommendations from the National Cancer Institute Physical Sciences - Oncology Centers Network based on the ATCC baseline protocol [1]. References [2] and [3] cited below have confirmed and updated the ATCC culturing condition based on the ATCC baseline protocol. In our laboratory, we have consistently used these protocols in our works without any visible signs of toxicity or alterations in cell morphology [4, 5]. We updated the Culture conditions section of the Material and Methods (Ln 587-594 in the revised manuscript) to accurately reflect the culture media and conditions used in the study.
References:
1. Homo sapiens ATCC HTB-26TM. https://www.atcc.org/Products/All/HTB-26.aspx. Accessed 3 January 2020.
2. Physical Sciences - Oncology Centers Network, Agus DB, Alexander JF, Arap W, et al:. A physical sciences network characterization of non-tumorigenic and metastatic cells. Sci Rep 2013, 3, 1449.
2
3. Kuhn NZ, Nagahara LA: Integrating Physical Sciences Perspectives in Cancer Research. Sci Transl Med 2013, 5(183):183fs14,1-3.
4. Shiau JY, Chang YQ, Nakagawa-Goto K, Lee KH, Shyur LF: Phytoagent deoxyelephantopin and its derivative inhibit triple negative breast cancer cell activity through ROS-mediated exosomal activity and protein functions. Front Pharmacol 2017, 8:398.
5. Nakagawa-Goto K, Chen JY, Cheng YT, et al: Novel sesquiterpene lactone analogues as potent anti-breast cancer agents. Mol Oncol. 2016;10(6):921–937.
